# Using Participatory Narrative Inquiry to Assess Experiences and Self-Experimentation with Diet Interventions in Inflammatory Bowel Disease Patients

**DOI:** 10.3390/nu16234027

**Published:** 2024-11-24

**Authors:** Celine Hos, Merel Tebbens, Tjitske Bezema, Jos A. Bosch, Aletta D. Kraneveld, Corinne E. G. M. Spooren, Marie Claire de Haas, Pieter C. F. Stokkers, Marjolijn Duijvestein, Gerd Bouma, Anje A. te Velde

**Affiliations:** 1Tytgat Institute for Liver and Intestinal Research, Amsterdam UMC, University of Amsterdam, Meibergdreef 69-71, 1105 BK Amsterdam, The Netherlands; 2Immunowell Foundation, 3947 NZ Langbroek, The Netherlands; 3Department of Psychology, University of Amsterdam, 1018 WS Amsterdam, The Netherlands; 4Department of Medical Psychology, Amsterdam UMC, University of Amsterdam, 1105 AZ Amsterdam, The Netherlands; 5Division of Pharmacology, Department of Pharmaceutical Sciences Utrecht Institute for Pharmaceutical Sciences, Utrecht University, 3584 CG Utrecht, The Netherlands; 6Department of Internal Medicine, School of Nutrition and Translational Research in Metabolism (NUTRIM), Maastricht University, 6200 MD Maastricht, The Netherlands; c.spooren@maastrichtuniversity.nl; 7Division of Gastroenterology-Hepatology, Maastricht University Medical Centre, 6229 HX Maastricht, The Netherlands; 8Department of Gastroenterology and Hepatology, OLVG West, 1061 AE Amsterdam, The Netherlands; 9Department of Gastroenterology, Radboud University Medical Center, 6525 GA Nijmegen, The Netherlands; 10Department of Gastroenterology and Hepatology, Amsterdam UMC, University of Amsterdam, 1081 HV Amsterdam, The Netherlands; 11Amsterdam Gastroenterology Endocrinology Metabolism (AGEM), 1105 AZ Amsterdam, The Netherlands

**Keywords:** inflammatory bowel disease, diet, lifestyle, quality of life, self-experimentation

## Abstract

Background and Aims: To improve quality of life (QoL), patients with inflammatory bowel diseases (Crohn’s disease and ulcerative colitis) often self-experiment with lifestyle changes such as dietary modifications. The nature (e.g., type of interventions, expectations, perceived efficacy) of these single-subject experiments has not been systematically investigated. Method: We used Participatory Narrative Inquiry (PNI), a structured qualitative method, to obtain information about these experiments through patient stories. Results: We demonstrate that PNI can be a method to collect and analyze IBD patient ideas and experiences regarding lifestyle and nutritional factors in a structured manner to reveal valuable insights for personal and scientific follow-up research. Patients report rest, (psychological) balance, and a change in diet when describing times when they experienced a better QoL. When focusing on diet, patients reported a considerable number of food products that were experienced as beneficial by one person but detrimental by another. Conclusions: PNI is a suitable method to obtain information about self-experimentation. An insight that was attained was that personalized (dietary) guidance that supports the individual is needed.

## 1. Introduction

Inflammatory bowel diseases (IBDs) are complex chronic inflammatory conditions that result in mucosal damage of the gastrointestinal tract, and include Crohn’s disease (CD) and ulcerative colitis (UC). Chronic inflammatory activity varies in different stages of the disease, including during the relapse and remission phases. Patients can suffer from symptoms such as bloody diarrhea, abdominal pain, weight loss, fatigue, and other complications, like fistula or abscess formation, which can lead to hospitalization and/or surgery. The physical impairments caused by IBD, their unpredictable course, the required lifelong medical surveillance, the lifelong medications, the treatment side effects, and the risk of malignancy all substantially contribute to a reduced quality of life (QoL) [1,2,3].

The treating physician determines the disease activity by biomarkers, imaging (bowel ultrasound, MRI), or endoscopy [4,5]. Patients themselves do not focus solely on their physical wellbeing but also on their mental and social health [6]. In addition, the inflammatory biomarkers might not correlate to the patient-perceived physical symptoms [7]. This gap between the doctor’s perception of the course of the disease and the patient’s more holistic perception of their wellbeing can lead to patients searching for interventions elsewhere. For instance, Hou et al. (2014) [8] reported a great desire among patients to receive dietary advice. Solid dietary advice based on clinical trials that focus on the effects of dietary intake on IBD is not available, because there are no large, well-designed, randomized clinical trials that support this, and it is questionable if this is feasible at all [9,10,11,12]. Therefore, patients may search on the Internet for alternative information resources. Many patients self-experiment with interventions involving lifestyle and dietary intake [13,14,15]. However, the results of these self-experiments are not systematically documented, and thus do not contribute to the current knowledge base.

So, how can the results and experiences of patient self-experiments be systematically investigated? In a previous paper from the Immunowell Foundation [16], we suggested that a way to integrate experience-based knowledge into scientific research is to use narratives to gain unexpected insights and leads in this complex domain. Narrative research is derived from the humanities and offers new methodologies for qualitative and quantitative studies of health and illness [17]. To deepen the impact of narratives in understanding patient experiences, clear and specific protocols are needed [18]. Participatory Narrative Inquiry (PNI) is a method to obtain such information through stories, and it consists of three components: participation, narration, and inquiry [19]. The first component of PNI research, participation, is the collaborative process where all parties that participate in the research process (patients, researchers, medical practitioners, and/or health organizations) are involved in designing questions and analyzing and interpreting the data. The narrative component of PNI research is focused on stories as an open way of collecting information rather than using questions that lead to pre-defined quantifiable answers. In addition, in PNI, the story is interpreted by the person telling his or her story [19]. The inquiry part refers to procuring new insights and directions. When the acquired data are analyzed, a deeper level of understanding of the study topic can be obtained, which can be used in decision-making, communication, and interventions. Research outcomes can give future directions for both personal and scientific research. PNI research is used as a means to achieve this goal rather than to answer specific hypotheses. We have used PNI to collect and analyze IBD patient experiences about lifestyle and nutritional factors in a structured manner to reveal valuable insights for personal and scientific follow-up research.

The purpose of this article is to illustrate that the PNI method can help to shed light on which aspects patients perceive to be associated with their QoL and what they do to improve it. This method may spur scientific research on lifestyle interventions that patients perceive to be helpful. The present study applied this method in two surveys involving over 250 patients with CD or UC, exploring the personal experiences that were (1) associated with a better QoL via the project named “Gut Feeling” and a resulting second project, or (2) that reported dietary intake that influenced QoL via “The Dietary Inquiry” project. The results of both projects were discussed and can form a basis to define hypotheses for further research regarding how to increase the QoL of IBD patients.

## 2. Materials and Methods

### 2.1. General Study Design

The idea to use the Participatory Narrative Inquiry (PNI) method to study the wellbeing of IBD patients originates from experiential experts within the Immunowell community. PNI is carried out in cycles (Figure 1) following six specific steps, and then returns to a new cycle if necessary and/or meaningful [19]. Patient participation was included in the following phases: 1/Design of questions, 4/Sense-making, and 5/Insights, hypotheses, and interventions.

#### 2.1.1. Design of Questions

The first phase is to design a mix of questions to elicit a *story about a personal experience*. The question is designed to avoid a yes/no answer, a personal opinion, or mere reference to information elsewhere. To facilitate this, questions about a situation can be accompanied by the question ‘what happened?’, which often leads to an answer of personal experience. Questions were designed and tested in collaboration with a small (three-to-six-person) group of IBD patients, two gastroenterologists with a special interest in IBD, and two scientists. This is a suitable number to carry out the design-of-questions phase in projects like these [19].

The patients asked to clarify some of the questions so that it would be more clear what exactly was asked, the scientists added some of the meta-questions that would give more ‘exact’ focus/information about the stories that were going to be submitted, and the doctors added the specific symptoms patients could experience.

In both projects, patients could choose to answer one of four story-provoking questions. The questions could be answered in an open field in which the experience could be written. Meta-questions about the participant and about the story are presented in Table 1. In project Gut Feeling, we wanted to explore how IBD patients described what happened in times they felt relatively well. When setting up the questions, it turned out that some IBD patients found it difficult to think of situations when they felt ‘good’. Instead, they more vividly remembered the times when they felt bad. Therefore, we added a question where they were asked to ‘remember a time when they felt bad, followed by a time they felt good’. For some people, this turned out to be a better incentive to write their story.

In addition to the story-provoking questions, a set of questions about the stories is determined. Participants reflect on their own story by giving it a title and via questions such as ‘is your story about…’ or ‘how much better did you feel in your story’. Finally, questions about the participants themselves (such as age, gender, and diagnosed condition) are included as meta-data.

##### Questions in Project Gut Feeling

Story-provoking questions are presented below:
1. Do you remember a time where you felt surprisingly good? What happened during that time? 2. Could you tell something about a situation in which your symptoms were surprisingly less than usual? What happened during that time? 3. Do you remember times where you changed something in your life that made you feel better, what happened? 4. Do you want to tell about a situation in which your symptoms first increased and subsequently decreased, what happened? 

In The Dietary Inquiry, the patient was asked to tell a story about their experience with food and drinks or with a change of diet. They were also asked to specify whether the story included an improvement or deterioration and to what category the dietary products mentioned in the story belong (gluten, refined sugars, sugars, saturated fats, unsaturated fats, fibers, dairy, vegetables, fruit, meat, fish, unprocessed foods, alcohol, coffee, probiotics, prebiotics, curcumin, Δ9-tetrahydrocannabinol/cannabidiol (THC/CBD) oil, other); what symptoms where described in the story (abdominal pain, increase in stool frequency, loose stools, rectal blood loss, decreased appetite, weight loss, general wellbeing, fatigue, other); in what timeframe the effect of the food occurred; and, lastly, whether their medication was changed in the particular time about which they wrote, and if so, what that change consisted of (see Table 2). Meta-data about the participants included age, gender, disease type, and comorbidities; these are also listed in Table 2.

##### Questions in the Dietary Inquiry

Story-provoking questions are presented below:
1. What was the best experience you have ever had regarding the effect of food and drinks on your physical condition and well-being? What food and drink were involved? 2. What was the worst experience you have ever had regarding the effect of food and drinks on your physical condition and well-being? What food and drinks were involved? 3. Do you remember a change in your diet (including drinks) which made you feel better. What happened around that time and of what consisted this change? 4. Do you remember a change in your diet (including drinks) which made you feel worse. What happened around that time and of what consisted this change? 

In order to come to a set of meaningful questions, in this phase all questions are tested on a smaller group of participants and then adjusted according to feedback from both patients and medical practitioners/researchers, after which the actual story collecting starts.

#### 2.1.2. Collection of Stories and Meta-Data

Due to the sensitive nature of patient stories and the fact that it is necessary for a meaningful result that people share fully and openly, the collection of the stories is performed anonymously. The stories and meta-data are collected via an online questionnaire form and cannot be traced back to the participant.

Participants were recruited via social media and patient organizations (Crohn & Colitis NL and the Immunowell foundation (www.immunowell.com). Stories from IBD patients were obtained via two online platforms: Gut Feeling: www.hiddenhealthsolutions.com and The Dietary Inquiry: www.voedinghelpt.nl. Data collection was performed anonymously. In addition to participating, patients could subscribe to a mailing list (separate from the questionnaire) for possible participation in the sense-making sessions and to be informed about the results of the projects.

#### 2.1.3. Meta-Analysis

In the meta-analysis phase, both the stories and meta-data (i.e., data that describe other data, see http://www.storycoloredglasses.com/p/pni-justified.html) are explored in order to prepare for the next PNI step, sense-making. With specific PNI software (version 1.4.0), such as NarraFirma^TM^ (version 1.4.0), the collected meta-data can be analyzed to extract trends, similarities, and exceptions. At the same time, all stories need to be pre-read and compared to the meta-data analysis. This leads to the script and agenda for the sense-making phase.

Meta-data were analyzed for patterns and trends using the open-source software NarraFirma^TM^, which is designed to carry out and support all steps of the PNI cycle.

For both projects, all stories were pre-read. Submissions from non-IBD patients and non-narrative submissions (i.e., not describing a personal experience related to the questions) were excluded. The meta-data can be used to gain a more detailed insight from the patterns, trends, and insights that emerge from the stories.

In the project Gut Feeling, factors and aspects in the stories that patients associated with wellbeing were counted. The results of this were stored in a mind map that served as a comparison backup for the sense-making session (see Appendix A). Further analysis of the meta-data was performed after the sense-making session in order to provide more detail for the overall trends and patterns in the results.

Factor occurrence of specific food products and drinks and their perceived effect on wellbeing were noted in The Dietary Inquiry project. The results were depicted in a word cloud with percentages, along with the possible meaning of certain patterns and trends in these results, to be discussed in the sense-making session. A literature study was carried out after the sense-making session to compare the existing literature with the overall trends and patterns in the results of the sense-making session.

#### 2.1.4. Sense-Making

Sense-making is the second co-creative phase (after the design of questions) where patients work together with scientists, medical practitioners, and/or health organizations to make sense of the stories and the meta-analysis. This is the key phase of the method where new insights emerge. Its objective is to construct the ‘overall story of all collected stories’ in terms of the following questions: What is the overall message of these stories? Which patterns emerge? What do we learn from these stories?

Sense-making was conducted by physically coming together with a subgroup of the participants (patients), scientists, medical practitioners, and/or representatives of health organizations. Together, they go through all the stories, factoring each story’s individual message and putting this together. A group with a total of ten to fifteen people is suitable for projects like these [19].

For the Gut Feeling project, the stories were analyzed in a sense-making session in collaboration with a group of IBD patients, IBD doctors, and researchers. The IBD patients were participants that subscribed to the mailing list and responded to the invitation for the sense-making session. The group in the sense-making session consisted of seven IBD patients, two IBD doctors (GB and PS), and three researchers experienced in IBD. All stories with included meta-data were read and discussed in subgroups, and a mind map (Appendix A) was created with patterns and trends.

The group in the sense-making session for The Dietary Inquiry project consisted of eleven IBD patients, one representative of an IBD patient organization (Crohn & Colitis NL), one IBD doctor (GB), and three researchers experienced in IBD. The results of the factor occurrence counting of specific food products and drinks and their perceived effects on wellbeing were presented and discussed. All stories that included meta-data were read and discussed in subgroups to create an overview of stories that either described effects of avoiding or taking single food products, or effects of avoiding or taking more than a single food product (or groups of food products).

To conclude, the group put together three messages from a combination of all the stories:A message for IBD patients;A message for gastroenterologists, specialized nurses, and dietitians;A message for researchers.

These messages are available on the website of the project (www.Voedinghelpt.nl).

#### 2.1.5. Insights, Hypotheses, and Interventions

In this phase, the insights and hypotheses from the sense-making phase are formulated as clearly as possible. These insights and hypotheses can be input for a subsequent PNI cycle where new stories are collected, as was done in our case; The Dietary Intake was a result of one of the outcomes from project Gut Feeling, and a new PNI cycle was started. In addition, these insights and hypotheses can be used to create or give advice about specific interventions related to the subject. From the sense-making sessions, several depictions were made that represented their results textually as well as visually.

#### 2.1.6. Return Insights

An important step in the PNI method is the return of insights to the patients who shared their stories. This step is essential because only after receiving the insights and hypotheses from the collective stories are participants able to learn from their fellow patients, and thus, only then have they fully participated in the PNI cycle as it was meant to function. Patients can then start (or continue) their own learning process.

The return of insights is performed using websites, magazines, social media, mailings, and/or meetings related to the subject that was explored.

An overview of the most prevalent lifestyle factors associated with wellbeing and advice with regard to further research resulting from the Gut Feeling project were presented (in Dutch) on a website [https://www.hiddenhealthsolutions.com/welkom-bij-project-onderbuikgevoel/patronen-in-de-verhalen-van-project-onderbuikgevoel], on social media, and published in the magazine of Crohn & Colitis NL.

Concerning The Dietary Inquiry project, an overview of food products and their perceived effects on wellbeing, the three messages from the stories, and advice with regard to further research were presented (in Dutch) on a website [https://www.voedinghelpt.nl/], to the Dutch Digestive Disease Foundation, on social media, and published in the magazine of Crohn & Colitis NL. Much attention was given to the fact that the method does not deliver proof but rather inspiration and new hypotheses and should be read as such.

## 3. Results

### 3.1. Gut Feeling

#### 3.1.1. Meta-Data Demographics

A total of 78 stories were collected. Demographics revealed that participants with both CD and UC were equally represented (51% and 46%, respectively; 3% of people did not have a conclusive diagnosis from their physician). The ages of the participants ranged from 20 to 70 years or older, with 76% of participants being 20–60 years of age. Females represented 75% of the participants.

#### 3.1.2. Prevalence of Factors in Patient Experiences

After all the stories were collected, a sense-making session was organized in order to comprise an overview of the factors that were mentioned most frequently.

After the sense-making session, an extra meta-analysis was performed to gain more insight into how specific lifestyle factors influence symptoms in relation to other lifestyle factors. While the outcome from the sense-making session only shows the relative prevalence of the named factors (Table 3), in many instances the patients mentioned multiple factors per story (Figure 2).

To gain a better understanding of the relation between different factors and symptoms, we looked at how many factors were mentioned in a story related to the most mentioned individual factors. In 56% of the stories, only one factor was mentioned. A total of 31% of the stories mentioned two factors, and 13% of the stories mentioned three or more. Figure 2 shows that when only one factor was mentioned, rest–balance and diet were mentioned most often, followed by medical intervention. When three or more factors were mentioned per story, the most regularly mentioned factors were rest–balance, physical activity, and diet. Physical activity was mentioned almost exclusively in combination with other factors. The opposite was seen for medical intervention, social support, and autonomy, which were barely mentioned in combination with three or more factors. Rest–balance and diet were the most consistent factors, both when mentioned as a single factor or in combination with one or multiple other factors.

#### 3.1.3. Meta-Data Patient Interpretation

Using the Narrafirma^TM^ software, associations between different meta-questions can be made. In this example, we addressed (1) the question regarding which lifestyle factors were changed in patients’ narratives and (2) which IBD symptoms were decreased. Figure 3 shows the correlation between how many times patients mentioned several changed lifestyle factors and which disease symptoms were decreased.

Figure 3 reveals that the change in lifestyle factor mentioned most frequently was the level of stress experienced by the patient. A reduction in stress levels was most mentioned in combination with a decrease in fatigue. To a lesser extent, the reduction in stress levels showed a relationship with a positive effect on abdominal pain, defecation problems, and depressive feelings. Other lifestyle factors mentioned regularly were change in diet, social life, self-esteem, “following your heart”, and physical activity. Following your heart is actually an interesting meta-data question because it was meant as a life choice matter (if someone was doing the things in his/her life that really felt close to their heart). However, most of the answers related to this specific meta-data category had another meaning, namely, ‘I followed my heart and tried several changes in my life(style) even though the doctor said it probably wouldn’t make any difference’ (while they felt it did).

Similar to changes in stress, the factors personal value, “following your heart”, and physical activity mostly affected the symptom fatigue. A change in diet was mostly mentioned as affecting abdominal pain, but it also had an effect on defecation problems and fatigue. Support from social life was equally associated with reductions in fatigue, abdominal pain, defecation problems, and depressive feelings. Thus, one might want to conclude from this grid that stress, followed by “following your heart”, self-esteem, and social life, are the most pronounced factors influencing QoL. Change in diet, described by 34% of the participating patients, did follow the aforementioned factors affecting core IBD symptoms (abdominal pain and defecation problems) and fatigue. Combined with the great desire among IBD patients to receive dietary advice, these data led us to start the project The Dietary Inquiry.

#### 3.1.4. Change in Diet and IBD

A change in diet was the second most mentioned factor related to better wellbeing (Table 3). In most stories, very little was mentioned regarding the specific food products or drinks that had an effect. Furthermore, ample research is available on the effects of dietary intake on the QoL of IBD patients. In order to create useful insights or hypotheses for further research, more information was needed on the effects of specific food products and drinks.

Therefore, it was necessary to carry out a second PNI cycle in which IBD patients were asked to describe their experiences with the effects of food products and/or drinks on their QoL.

### 3.2. The Dietary Inquiry

#### 3.2.1. Meta-Data Demographics

A total of 203 stories were collected. Participants with both CD and UC were equally represented (CD 51%, UC 47%, 2% of people did not have a conclusive diagnosis from their physician) and 80% of the participants were female. A total of 92% of the participants were between the ages of 20 and 60 years old. The patients selected one out of the four questions (1: 29%, 2: 29.5%, 3: 29.5%, and 4: 12%), indicating the relevance of using different story-provoking questions.

#### 3.2.2. Prevalence of Dietary Factors in Patient Experiences

After analysis of the stories gathered in The Dietary Inquiry project, a list of the top five food and drink products emerged. The top five list includes food and drink products patients avoid/reduce to increase their QoL. Those food products are dairy, sugars, gluten, meat, and alcohol, which were mentioned in 36%, 31%, 26%, 19%, and 17% of the stories, respectively (Figure 4A). Other food and drink products patients said to eat less of to increase QoL are food additives (E-numbers), coffee, bread and yeast, nightshade vegetables, spicy food and spices/herbs, fats and oils, aerated drinks, onion varieties, mushrooms, cabbage varieties, and chocolate. The top five food and drink products patients eat more of to increase their QoL are vegetables, homemade foods (not industrialized ultra-processed food), fruits, water (including herbal tea), and nuts, which were mentioned in 23%, 14%, 11%, 9%, and 8% of the stories, respectively (see Figure 4B). Other food products patients stated they eat more of to increase their QoL are fermented products, oats, and spelt instead of wheat. The food products mentioned in less than 5% of the stories can be found in Appendix A.

In addition to the top five food products to be avoided and increased, the sense-making session pointed out that half of the participants mentioned having only made an adaptation to reduce or increase a single type of food (or drink) product. They mention, for instance, that they avoid taking it because they noticed it worsened their symptoms. The other half carried out a more elaborate change in diet by changing groups of products.

#### 3.2.3. Participant Quotes

“Don’t fill but feed: The exclusion of gluten, sugar and dairy has only brought good things. Much less flare-ups and stomach aches. No soft drinks anyway, prepare meals fresh every day. No food in packages, bags, etc. It keeps me busy, but it certainly has a positive effect”.

“Eat fresh and unprocessed foods: Since I eat less lactose and gluten and eat as much fresh and unprocessed food as possible, my pain and fatigue complaints have decreased by at least half. I now know that I can’t handle ready-made food, so I make a lot myself”.

“Improve your illness and yourself through nutrition: […..} My doctor was very surprised, my secret = Healthy natural unprocessed food! It mostly surprised me doctors say so little about nutrition, they give you medicines in the hope of getting/keeping everything under control. But that’s not necessary, as long as you watch what you eat. Seriously, do something with this information, it could help so many people. This is why I am writing this post. I am not a dietitian or a nutritionist, I am a 16 year old girl who was looking for a solution to improve her life with this disease”.

Overview of some other titles of stories: ”Food can change everything for you”, “The success formula of “IBD, eet je mee” (a Dutch IBD diet program), “Thanks to the Paleo diet, medication and complaints free for 4 years now”, “Do not eat!”, “Nutrition helps to calm intestines”, “Search for the best possible quality of life through nutrition”, “Huge change through eating differently”, “Healthy gut: a combination of healthy food and common sense”, “Doctors: also think about a different diet”, “Dairy-free and no alcohol”, “Listen to your body, you really know better than others”, etc.

## 4. Discussion

Being chronically ill is life changing. By giving patients the opportunity to tell their stories, everyone involved in the project (including doctors, researchers, and the participants themselves) gains a deeper understanding of the different aspects of the(ir) disease. For IBD patients, being diagnosed as ‘in remission’ does not guarantee a good QoL, and being diagnosed as having a flare does not always mean a patient feels very ill. Certain models that look at health already exist, where health is approached in an all-inclusive way, looking at a variety of factors such as stress, lifestyle influences, and mental and physical health. In this view on health, disease or illness is presented as a health determinant and not so much as a main focus point [20,21]. Chronically ill patients might have constitutionally different outcomes on the proposed scales that determine health when compared to non-diagnosed individuals. On the other hand, although clinical scores might not give a full representation of how the patient is feeling (in other words, the perception of QoL), they do indicate the severity of the disease on a medical and physical level [22]. It would be an interesting proposition to combine our findings of the Gut Feeling study on the various categories of what patients find beneficial for their QoL with existing models of health and with current clinical outcomes to generate a more holistic view on a patient’s state of wellbeing [23].

In The Dietary Inquiry study, we observed that the effects of dietary changes are highly personal. Some food products were mentioned by some patients to increase their wellbeing, while for others these products decreased their wellbeing (see Figure 4 and Appendix A). This may be one of the reasons the effects of RCTs analyzing standardized dietary interventions are uncertain and do not give clear answers [10,24,25]. In addition, the standard methodology including a double-blind setting is not possible in dietary research. Most reviews and meta-analyses conclude that there is insufficient evidence for a specific diet to promote induction or maintenance of clinical remission [3,10,26,27]. Fortunately, things have recently changed, and in a recent open-label pilot study with a Crohn’s disease exclusion diet, effective induction of remission in mild-to-moderate naïve CD patients was demonstrated [28,29], and in an IBD cohort study, an association of the dietary pattern with the occurrence of flares was found [30]. In addition, clinicians regularly observe that patients change their diet based on perceived associations with their symptoms or disease relapses [31,32]. Nevertheless, based on possible effects on gut health, advice to reduce ultra-processed foods in the diets of IBD patients is given in various papers [33,34,35].

Our study also highlights the importance of the development of a personalized approach. Therefore, we propose a new strategy to accommodate this. The strategy should include the use of PNI on a personal level. If a patient wants to experiment with their diet (or other lifestyle factors), they will be able to start a self-experiment with the guidance of their doctor and dietician. This guidance is highly recommended to prevent malnutrition. Their findings will be reported in the form of stories and analyzed using PNI. In this way, the patient can self-experiment in a safe and guided manner.

People with IBD share stories about how they changed their lifestyle and how it affected their quality of life. They thus form a source of knowledge and awareness for others (doctors, dietitians, nurses, lifestyle coaches, people with IBD and their loved ones, etc.). In the short term, this can provide inspiration for other people with IBD who want to start their dietary and/or other lifestyle interventions.

This research is a form of an empirical approach to self-structured observation. We now know that healthy food, stress reduction, better sleep, and exercise are important for a patient’s quality of life, but knowing at an individual level is not yet doing. Information from people with IBD who experience the effects of a change in their lifestyle can be an inspiration to ‘do it’ [36].

If people decide for themselves how they conduct research, they will possibly conduct it much better; moreover, if lifestyle changes are not imposed but are based on the experiences of others, it may be easier to convince people that it can be achieved.

### 4.1. Lessons Learned for Future Use of PNI

Even though the PNI method has been applied in many areas of expertise and thereby has acquired a status of its own [19], to our knowledge, it has not been applied in studies (of quality of life) in patients. We demonstrate that it can provide information on how IBD patients experience the effects of lifestyle factors on their QoL and the course of their disease. In carrying out the two presented projects, we encountered several aspects that can be taken into account when starting new PNI-based projects.

First, we observed that the stories themselves, rather than the meta-data, were leading and gave definite explanations of the results. It is probably difficult for participants to both (1) write down their story and (2) take a step back and answer questions about their own story (as if they were a researcher inquiring about their own story). It might be better to not add too many questions about the story and to formulate their meaning in a simple way.

Second, the sense-making sessions with patients, doctors, and researchers are intensive, and patients who are chronically ill often have limited energy, so attending a full session of three hours may be too exhausting for them. It would be better for the sense-making sessions to be divided into several shorter sessions, for instance, with fewer participants per session. On the other hand, doctors barely have the time to attend sense-making sessions. It is best to gain commitment from the doctors in the project team to either attend more sessions or to add enough doctors to the project team so that every session can be attended.

Finally, it is important to keep in mind that PNI is a method to generate new insights and possibly identify a possible relationship between lifestyle factors and QoL. Several limitations of the current results should be noted. First of all, this study might be subject to population bias. The participants in this research are anonymous IBD patients. They are also patients who are willing and able to share their experiences. The patients who are not willing or able to share their stories are left out, and therefore their experiences are unaccounted for. Secondly, narrative research provides a substantial body of data and is therefore prone to over-interpretation and finding irrelevant correlations [17,37]. The concept of validity in narrative research is slightly different than in evidence-based medicine [38]. Validity in narrative research is determined by the context of the research; the outcomes have to serve the research goal. Every story is as valid as the whole body of data together. To fully understand the impact of the research, a proper assessment is essential. Although we also used quantitative methods to analyze our results, the interpretation thereof is still susceptible to bias. To reduce this risk, we looked at the unprocessed data (the stories) and the analyzed data (the top fives) during the sense-making session with IBD patients, gastroenterologists, and the researchers involved in this study. By comparing the interpretations from people with multiple perspectives, a more unbiased interpretation can be made. This constant comparison and feedback by participants is a form of respondent validation and enables us to re-evaluate the data and interpretations [37]. Thirdly, a story may contain more than one factor, that is, more than one change in lifestyle. It was not possible to assess whether more changes also induced greater improvements in QoL. In addition, we did not inquire about what lifestyle or specific dietary changes led to a particular decrease in symptoms. The results of this study are therefore only an indication that these changes might play a role in the course of the disease, and the result, as stated before, should be further investigated in more controlled and quantitative settings.

### 4.2. Messages That Emerged from the Stories

In the sense-making session, several messages clearly emerged from the stories:The most frequent factors that were mentioned by participants when describing times when they experienced a better QoL were rest and (psychological) balance and a change in diet.Participants described the effects of dietary changes on the QoL they experienced in terms of a reduction in symptoms such as abdominal pain, fatigue, and defecation problems.The message to researchers is to conduct more research on the effects of the top five food products on IBD symptoms. When the possible effects on IBD symptoms are known, a patient, together with a dietician, can find an individual diet that suits his or her needs while still taking in sufficient essential nutrients and not overdosing on others.A considerable number of food products were experienced as beneficial by one person and detrimental by another. This is shown in Appendix A, and results in the message that personalized dietary guidance is needed that supports the individual.

In conclusion, PNI is a suitable method to gain insights into which aspects IBD patients perceive to be associated with their QoL and what they do to improve it. The advantage of applying PNI in this manner is twofold. First, asking participants for answers in the form of narratives helps to discover aspects of QoL that are proposed by patients themselves instead of by researchers and medical doctors. In addition, aspects that the patient may not even realize to be important emerge when writing their story. This elicits more insight for each individual participant and, at the same time, makes the overall message from all the narratives together valuable for patients and doctors because it helps them to better understand each other. In addition, the value for researchers is to obtain new hypotheses, and the value for the patient population is in learning from each other’s experiences. Secondly, IBD patients described what they do themselves to reduce IBD symptoms and in which situations they felt better. The results of both the Gut Feeling and The Dietary Inquiry projects showed evident patterns of what factors IBD patients experience that are associated with their QoL, providing a solid basis to build on in future research. New research on dietary habits and IBD can focus on the nutritional components in the top five food products. More specifically, this can include clinical research on those nutritional components and their effects on QoL, including perceived symptoms, instead of (or in addition to) trying to prove or disprove a direct therapeutical effect in terms of IBD disease activity. When the possible effects on QoL are known, a patient, under the guidance of a dietician and a doctor, can find a personalized diet in a safe manner [39]. PNI can be applied to accommodate guidance for the individual patient. Both Gut Feeling and The Dietary Inquiry indicate that many IBD patients did change their diets (and/or other lifestyle aspects) to improve their QoL. Patients and their doctors should be aware that self-experimenting with their diet can lead to malnutrition. Therefore, we recommend addressing ways to improve the diet in the office of the medical team to search for ways to increase the QoL. In addition, based on our findings, we also recommend that IBD patients and their doctors should pay attention to the possible effects of rest and balance on the course of the disease and the possible improvement of QoL. Patients’ PNI stories could lead the way in the treatment of IBD and other chronic diseases [40].

## 5. Conclusions

Because IBD, like many other chronic conditions, is a complex disease, we must find a way to deal with it. This complexity can either be reduced or we can try to manage it [41]. Management can be achieved through learning, then adapting, and finally emerging. Extracting patterns from complex data can, as we have demonstrated here, be achieved by questioning practices, working narratively, and telling stories. PNI is an extremely suitable method for acquiring this implicit experiential knowledge, which is crucial for managing complexity and obtaining information on how to improve the QoL of patients with chronic conditions such as IBD.

## Figures and Tables

**Figure 1 nutrients-16-04027-f001:**
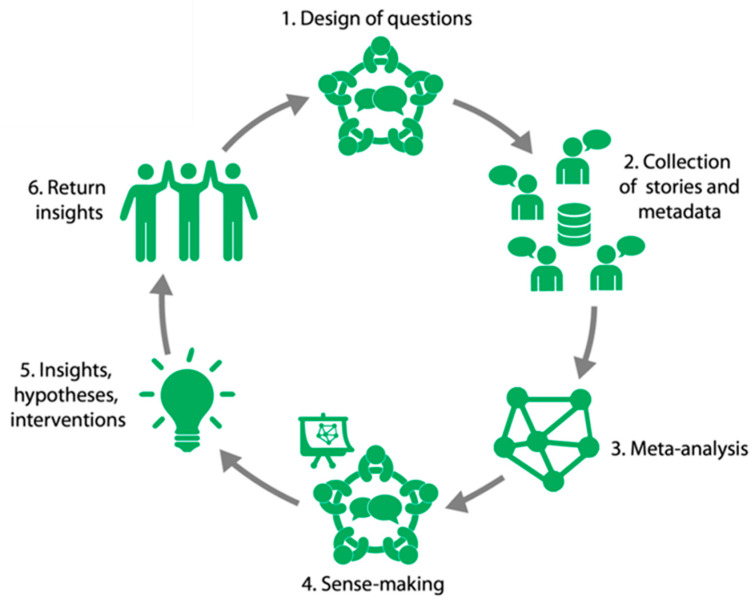
Six steps of the PNI cycle that cover the three features of PNI: participation, collection of narratives, and the generation of insights and hypotheses (inquiry).

**Figure 2 nutrients-16-04027-f002:**
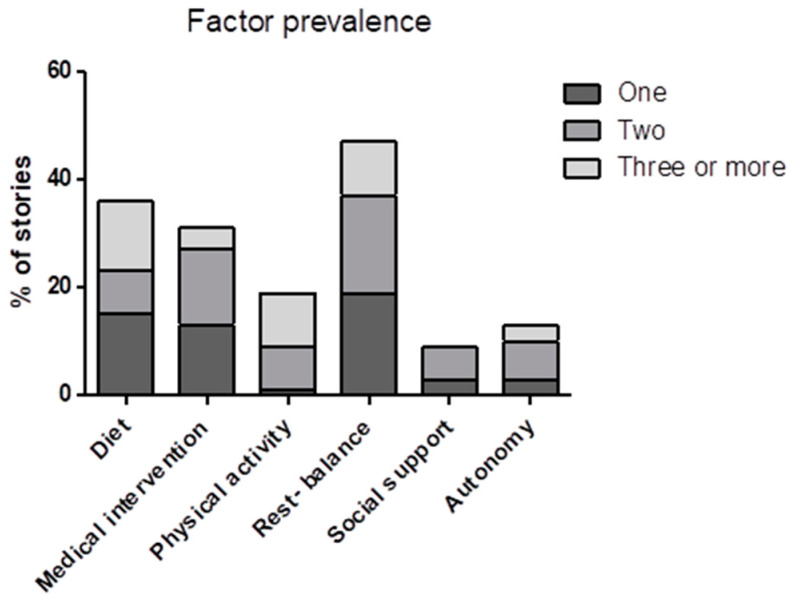
Representation of the results from the sense-making session indicating how often a factor was mentioned as an independent factor or in combination with one or multiple other factors per story.

**Figure 3 nutrients-16-04027-f003:**
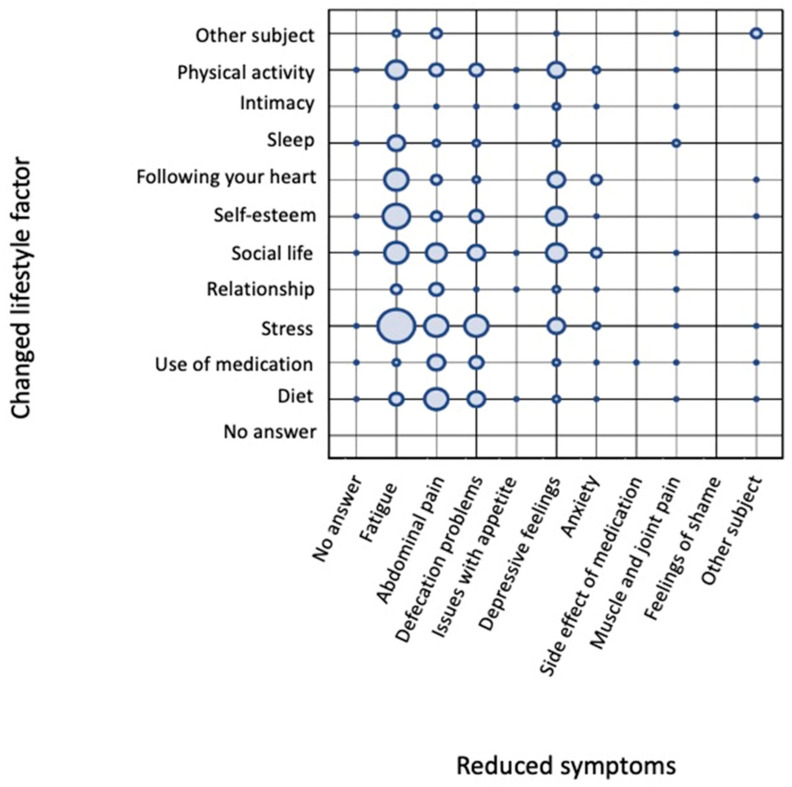
Grid of correlations between changed lifestyle factors and decreased symptoms. Lifestyle factors are presented on the y-axis and reveal that stress was mentioned most often, followed by social life, personal value, physical activity, diet, “following your heart”, use of medication, sleep, “other”, and intimacy. The affected disease symptoms are shown on the x-axis, with a decrease in fatigue mentioned most frequently, followed by stomachache, defecation problems, and depressive feelings. The circles represent a relative, rather than quantitative, means of evaluating how often specific subject–answer combinations occurred in relation to each other. This approach adds an additional dimension for evaluating the stories and formulating hypotheses.

**Figure 4 nutrients-16-04027-f004:**
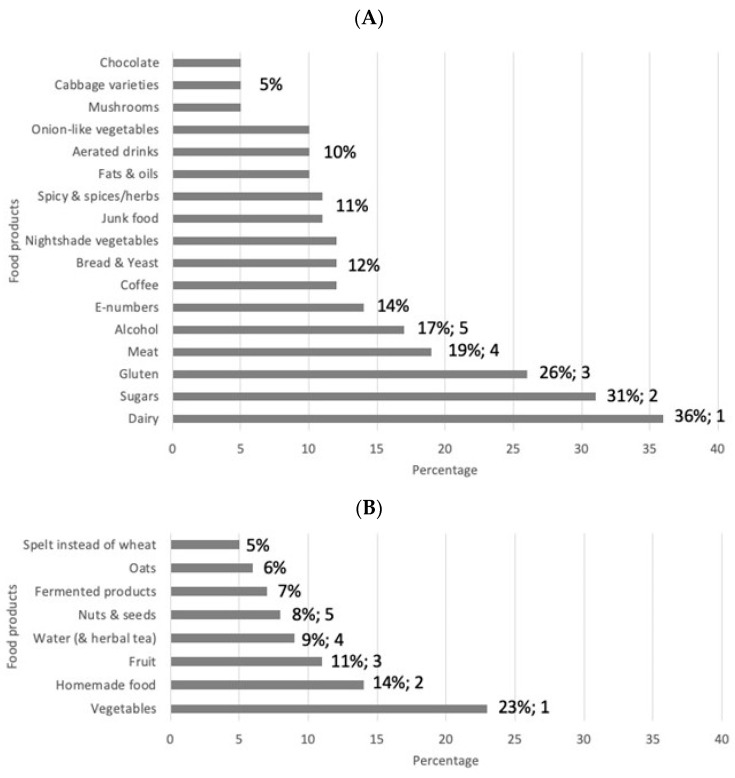
Diagram showing the relative frequency of food products mentioned by patients to AVOID/REDUCE (**A**) or EAT MORE OF (**B**) to increase their quality of life. The numbers indicate the percentages above 5%. E-numbers are codes for food additives that have been assessed for use within the European Union (the “E” prefix stands for “Europe”).

**Table 1 nutrients-16-04027-t001:** Meta-questions in project Gut Feeling.

Questions About the Participant	Questions About the Story
Age	Timeframe in which the effect was observed
Gender	Symptom(s) described
Disease type	Factors that changed
Habitat	Rural or urban
Use of medication	Yes/no

**Table 2 nutrients-16-04027-t002:** Meta-questions in project The Dietary Inquiry.

Questions About Participant	Questions About the Story
Age	Timeframe in which the effect of the food occurred
Gender	Symptom(s) described
Disease type	Dietary factors that changed
Comorbidities	Whether it contained improvement or deterioration
	Category of the food products described
	Whether medication was changed; if yes, what the change consisted of

**Table 3 nutrients-16-04027-t003:** The most frequent factors that were mentioned by patients when describing times when they experienced a better QoL (starting with the most frequent).

1. Rest and psychological balance (45%)
2. Change in diet (34%)
3. Medical intervention (via treating physician) (32%)
4. More physical activity (19%)
5. Patient autonomy (12%)
6. Social support (8%)

## Data Availability

Data are mini-narratives and are not presented in a standard data format, but are available for analyses upon request. The data are not publicly available due to the format.

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
