# Peer review of "Using Participatory Narrative Inquiry to Assess Experiences and Self-Experimentation with Diet Interventions in Inflammatory Bowel Disease Patients"

_nutrients, 2024, doi:10.3390/nu16234027_

Round 1
Reviewer 1 Report
Comments and Suggestions for Authors
The manuscript describes the participatory narrative inquiry method to assess experiences and self-experimentation: illustrated in inflammatory Bowel Disease patients. The topic is relevant to the aim and scope of the Nutrients. The manuscript is well written and easy to follow. However, this manuscript needs to address the below comments:
1. Questions were raised as Table 1, but any analysis was not shown in terms of the questions. Does no description for the analysis indicate that no effect of the questions was found?
2. The article title is “The Participatory Narrative Inquiry method to assess experiences and self-experimentation: illustrated in Inflammatory Bowel Disease patients”. However, the assessment is mostly about dietary inquiry. Please consider again the title matched to the content of this manuscript.
3. In Figure 3, what is the quantitative definition of the circle size? Please declare the definition first.
4. More explanation is necessary to make a systematic correlation between the lessons learned for future use of PNI and the messages acquired from the stories.
5. It is known that the Nutrients journal has a manuscript format including “Conclusions” section. Please modify this manuscript in such a way that it has a separate “Conclusions” section. So, the format can be kept.
Author Response
The first reviewer asked to address the following comments:
- Questions were raised as Table 1, but any analysis was not shown in terms of the questions. Does no description for the analysis indicate that no effect of the questions was found?
The questions in Table 1 are meta-questions and, by definition, broad because, in advance, we did not know what could be important. They are meant to provide some insight into the population that would respond to the story-provoking questions. We analyzed part of the questions related to the stories about symptoms described and factors that changed, as shown in Figure 3, to demonstrate what can be done with this data. The stories about the dietary interventions were so rich that we focused further on this aspect for analysis.
- The article title is “The Participatory Narrative Inquiry method to assess experiences and self-experimentation: illustrated in Inflammatory Bowel Disease patients”. However, the assessment is mostly about dietary inquiry. Please consider again the title matched to the content of this manuscript.
This is a good suggestion and we changed the title into: “Using Participatory Narrative Inquiry to assess experiences and self-experimentation with diet interventions in Inflammatory Bowel Disease patients”. We hope that this is a better match.
- In Figure 3, what is the quantitative definition of the circle size? Please declare the definition first.
This is a valid point and we added this information to the legends of figure 3: “The circles are defined as a relative, rather than a quantitative, means of evaluating how often specific subject-answer combinations occur in relation to each other. This approach adds an additional dimension for evaluating the stories and formulating hypotheses”.
4. More explanation is necessary to make a systematic correlation between the lessons learned for future use of PNI and the messages acquired from the stories.
5. It is known that the Nutrients journal has a manuscript format including “Conclusions” section. Please modify this manuscript in such a way that it has a separate “Conclusions” section. So, the format can be kept.
We combined these two comments and wrote a separate conclusion addressing point 4. “Because IBD, like many other chronic conditions, is a complex disease, we must find a way to deal with it. This complexity can either be reduced or we can try to manage it. Management can be achieved through learning, then adapting, and finally emerging. Extracting patterns from complex data can, as we have demonstrated here, be done by questioning practice, thus working narratively and telling stories. PNI is an extremely suitable method for acquiring this implicit experiential knowledge, which is crucial for managing complexity and obtaining information on how to improve the QoL of patients with chronic conditions such as IBD”.
Reviewer 2 Report
Comments and Suggestions for Authors
The manuscript entitled, The Participatory Narrative Inquiry method to assess experiences and self-experimentation: illustrated in Inflammatory Bowel Disease patients” aimed to propose a Participatory Narrative Inquiry to obtain information about self-experimentation and an insight into personalized (dietary) guidance to people suffering from inflammatory bowel diseases. Overall, this is an interesting study and which could be informative to people concentrated on public health or clinical dietetic working with critically ill patients.
To address this issue the authors collected over 203 stories and indicated food products which could reduce IBD symptoms and should be eaten in higher percentage to improve the quality of life. This study proposes a new approach to interviewing patients with chronical illness. The conclusions consistent with the evidence and arguments presented and they address the main question posed. The references and style of citations is accurate. This is an elegant study which summarizes the most important factors that influence lifestyle and change subjective symptoms of inflammatory bowel diseases.
Minor points:
The authors could increase the font type and do additional analysis with subgroups divided by sex and age to verify if there are any differences in preferences of verified groups.
Author Response
The authors could increase the font type and do additional analysis with subgroups divided by sex and age to verify if there are any differences in preferences of verified groups.
Increasing the font type could be a suggestion to the editing office.
The question is indeed whether sex and age are relevant for the outcomes related to dietary preferences. Due to the large variety of outcomes, the study is too small to draw conclusions about subgroups. A follow-up study could be set up with specific inclusion criteria to address this. This is exactly the purpose of PNI – hypothesis formation and serving as a source of knowledge for new follow-up research, which can be increasingly refined over time.
Changes made to the manuscript text:
Line 1-3: we changed the title according to comment reviewer 1.
Table 1: we removed the … after factors that changed.
Table 2: added dietary before factors that changed.
Line 39: added complex.
Line 133 and 148 removed bold.
Line 299-302 added explanation according to comment 3 reviewer 1.
Line 536-544 added conclusion section according to comment 4 and 5 of reviewer 1.
Line 658: reference 41 is added.
Round 2
Reviewer 1 Report
Comments and Suggestions for Authors
All issues have been addressed.